# Miniaturization of CRISPR/Cas12-Based DNA Sensor Array by Non-Contact Printing

**DOI:** 10.3390/mi15010144

**Published:** 2024-01-17

**Authors:** Hiroki Shigemori, Satoshi Fujita, Eiichi Tamiya, Hidenori Nagai

**Affiliations:** 1Advanced Photonics and Biosensing Open Innovation Laboratory (PhotoBIO-OIL), National Institute of Advanced Industrial Science and Technology (AIST), Photonics Center Osaka University, 2-1 Yamada-Oka, Suita 565-0871, Osaka, Japan; h.shigemori@aist.go.jp (H.S.); s-fujita@aist.go.jp (S.F.); tamiya.handai-oil@aist.go.jp (E.T.); 2Graduate School of Human Development and Environment, Kobe University, 3-11 Tsurukabuto, Nada-ku, Kobe 657-0011, Hyogo, Japan; 3Institute of Scientific and Industrial Research (SANKEN), Osaka University, 8-1 Mihogaoka, Ibaraki 567-0047, Osaka, Japan

**Keywords:** CRISPR/Cas12, collateral cleavage, protein array, genotyping, multiplex detection, printed biosensor, non-contact bioprinting, accumulated sensor array, miniaturization

## Abstract

DNA microarrays have been applied for comprehensive genotyping, but remain a drawback in complicated operations. As a solution, we previously reported the solid-phase collateral cleavage (SPCC) system based on the clustered regularly interspaced short palindromic repeat/CRISPR-associated protein 12 (CRISPR/Cas12). Surface-immobilized Cas12-CRISPR RNA (crRNA) can directly hybridize target double-stranded DNA (dsDNA) and subsequently produce a signal via the cleavage of single-stranded DNA (ssDNA) reporter immobilized on the same spot. Therefore, SPCC-based multiplex dsDNA detection can be performed easily. This study reports the miniaturization of SPCC-based spots patterned by a non-contact printer and its performance in comprehensive genotyping on a massively accumulated array. Initially, printing, immobilization, and washing processes of Cas12–crRNA were established to fabricate the non-contact-patterned SPCC-based sensor array. A target dsDNA concentration response was obtained based on the developed sensor array, even with a spot diameter of 0.64 ± 0.05 mm. Also, the limit of detection was 572 pM, 531 pM, and 3.04 nM with 40, 20, and 10 nL-printing of Cas12–crRNA, respectively. Furthermore, the sensor array specifically detected three dsDNA sequences in one-pot multiplexing; therefore, the feasibility of comprehensive genotyping was confirmed. These results demonstrate that our technology can be miniaturized as a CRISPR/Cas12-based microarray by using non-contact printing. In the future, the non-contact-patterned SPCC-based sensor array can be applied as an alternative tool to DNA microarrays.

## 1. Introduction

The recent surge in drug-resistant bacteria has heightened the demand for comprehensive pathogenic identification technology in the diagnosis of infectious disease and sepsis [1]. Specifically, any delay in initiating drug-administration in cases where sepsis patients exhibit shock symptoms directly correlates with a decrease in the survival rate [2]. Therefore, there is a pressing demand for the rapid identification of pathogens and their drug-susceptibility. In this context, the application of pathogenic genotyping technology has garnered attention, particularly because it eliminates the need for pathogen culturing over several days [3].

Nowadays, conventional genotyping technologies such as real-time polymerase chain reaction (PCR), DNA sequencing, and DNA microarrays are widely used. However, they still exhibit several drawbacks in clinical applications. For instance, real-time polymerase chain reaction (PCR) can detect as low as 1 copy/reaction of a target gene within 10 min using some ultrafast instruments [4,5]. Nevertheless, commercially available products cannot detect more than six target genes from a single reaction tube [6]. DNA sequencing directly reads the target gene sequence, although it takes from half a day to several days [7,8]. Meanwhile, DNA microarrays perform hybridization-based genotyping on a chip immobilized with various types of single-stranded DNA (ssDNA) sequences and is a comparatively high-throughput genotyping technology [9]. Thus, DNA microarray products like Verigene [10,11,12] and ePlex [13,14,15] are practically applied for pathogenic genotyping in respiratory tract infection and sepsis.

Despite the capability of DNA microarrays to simultaneously identify up to 300,000 target sequences [16], the process takes from 1.5 to 3 h to identify the target gene [10,11,12,13,14,15], owing to several operation steps, including gene amplification, target ssDNA synthesis from double-stranded DNA (dsDNA) amplicons, labeling, and washing [9,17]. Overall, conventional genotyping technologies face limitations concerning the coexistence of short detection times and the range of detectable target types. To overcome this limitation, the World Health Organization (WHO) advocates for the target product profile (TPP) for ideal genotyping technologies, demanding the simultaneous detection of fifteen or more pathogenic and drug-resistant genes within one hour [18].

Recently, the clustered regularly interspaced short palindromic repeat/CRISPR-associated protein 12 (CRISPR/Cas12) system has gained attention as a rapid nucleic acid detection tool. The complex of Cas 12 and CRISPR RNA (crRNA) directly hybridizes to complementary dsDNA sequences and activates the indiscriminate cleavage of surrounding ssDNA (collateral cleavage) [19]. Thus, the collateral cleavage-induced signal can be obtained by simply mixing dsDNA amplicons, Cas12–crRNA complex, and signaling probe-labeled ssDNA reporters [20,21,22,23].

While CRISPR/Cas12 can rapidly detect target dsDNA, usually within one hour [24], through a straightforward process, a technical hurdle remains for comprehensive genotyping. Activated Cas12 cleaves ssDNA reporters regardless of their sequences, although the multiple detection of several dsDNA targets is difficult to perform in a single chamber [25]. To overcome this drawback, Combinatorial Arrayed Reactions for Multiplexed Evaluation of Nucleic acids (CARMEN) [26], microfluidic CARMEN [27], and Cas-Loaded Annotated Micro-Particles (CLAMP) [28] were introduced to separate the sample solution into individual collateral cleavage chambers based on droplets, microwells, and hydrogels. These approaches allow for the accumulation of 84.44–1367.17 chambers on 1 cm^2^ of chip, facilitating an increase in the number of detectable target genes [29]. However, they typically require four or more operations, resulting in longer detection times (two hours or longer).

Meanwhile, several research groups have attempted to separate the sample solution into multiple assay spots at the ends of microfluidic channels [30,31,32,33,34,35]. These can be used by dropping the sample solution into the inlet, resulting in a shorter detection time. However, the accumulation of spots is limited to a maximum of 2.13 spots on 1 cm^2^ of chip, owing to the long length of channels to prevent cross-collateral cleavage reactions among assay areas [29]. In addition, the required volume of the sample solution increases with the expansion of the assay area since all assay areas and microfluidic channels must be filled with the sample solution.

Recently, the combination of Cas12-subtypes [36] and Cas13-subtypes [37] was widely adopted for collateral cleavage-based one-pot multiplexing. The collateral cleavage substrate is different between Cas12 (ssDNA) and Cas13 (ssRNA), and the sequence of the substrate depends on the Cas subtype. Thus, multiplex PCR-like detection systems can be realized by classifying Cas types, reporter sequences, and probe types for different targets. However, this sequence specificity of ssDNA substance has only been confirmed in two subtypes of Cas12 [36] and four subtypes of Cas13 [37], severely limiting the amount of detectable target genes. Overall, previous attempts at collateral cleavage-based multiplexing still exhibited limitations in simultaneously achieving comprehensive genotyping and simple operation.

We previously reported a sensor array model based on the solid-phase collateral cleavage (SPCC) system as a novel approach for facile multiplexing in collateral cleavage-based dsDNA detection [29]. In this prototype, Cas12–crRNA and the fluorophore-labeled ssDNA reporter are immobilized on the surface of each spot, with the crRNA sequence of each spot set to be complementary to the target dsDNA sequence of that spot. Once the target dsDNA is captured on the Cas12–crRNA-immobilized spot, the activated Cas12 cleaves the ssDNA reporter immobilized on the same spot. Considering that the activated Cas12 is immobilized on the spot surface, it cannot cleave the ssDNA reporter on the other spots. Therefore, the target dsDNA sequence can be identified by the spot with decreased florescence. Unlike the aforementioned approaches for multiplexing, the SPCC sensor array prototype successfully identified mutations in two target dsDNA sequences in a single reaction well. Additionally, the operation process requires only hybridization (at least 20 min to detect 100 nM target dsDNA) and washing steps.

Despite its promising performance, the SPCC-based sensor array model has some drawbacks. Considering that the reagent of the SPCC spot is dispensed by a micropipette, the spot-accumulation capacity (5.88 spots/cm^2^) is still lower than that of the collateral cleavage in massively accumulated chambers [26,27,28] and conventional DNA microarrays [16]. Furthermore, this dispensing system consumes a large amount of reagent (2.5 µL/spot). Additionally, the SPCC-based spot is patterned by the sealing-based contact process, which is challenging to fabricate on a micro-structured surface, and the spot-alignment is not controlled owing to the manual operation.

Herein, we report the miniaturization of the non-contact printing-based patterning of SPCC-based spots for comprehensive genotyping (Figure 1). Recently, the applications of non-contact printing technology have expanded to include fabricating DNA microarrays [38,39] and other chemical- and bio-sensors [40,41], owing to a high density of sensing spots, accurate spot-alignment, and low reagent consumption. We anticipate that these advantages of miniaturization using non-contact printing will address the drawbacks of the previous SPCC prototype.

This study established non-contact printing, immobilization, and washing processes for the Cas12–crRNA complex to fabricate miniaturized micropatterns for SPCC-based sensors by using non-contact printing. We confirmed the dsDNA concentration response and the feasibility of multiplexing in the system. We expect that the non-contact-patterned SPCC-based sensor array may serve as an alternative genotyping tool to DNA microarrays in the future.

## 2. Materials and Methods

### 2.1. Materials and Reagents

Alt-R L.b. Cas12a (Cpf1) Ultra was purchased from Integrated DNA Technologies Inc. (Coralville, IA, USA). CRISPR RNA (crRNA) was purchased from Thermo Fisher Scientific Inc. (Waltham, MA, USA). pVenus-N1 plasmid was re-constructed by inducing point mutations from the pEGFP-N1 plasmid that was obtained from Clontech Laboratories, Inc. (Mountain View, CA, USA). PCR primers and the ssDNA reporter were purchased from Eurofins Genomics K.K. (Tokyo, Japan). All oligonucleotide sequences are listed in Appendix A. N-Hydroxysuccinimide (NHS), N-(3-dimethylaminopropyl)-N′-ethylcarbodiimide-hydrochloride (EDC), Nα,Nα-bis(carboxymethyl)-l-lysine hydrate (NTA), and ethanolamine hydrochloride were purchased from Sigma-Aldrich Co. LLC (St. Louis, MO, USA). The 2-(N-Morpholino)ethanesulfonic acid (MES) used in this study was purchased from Dojindo Molecular Technologies Inc. (Kumamoto, Japan). Sodium hydroxide, sodium dihydrogenphosphate dihydrate, disodium hydrogen phosphate dodecahydrate, and nickel(II) chloride were purchased from Fujifilm Wako Pure Chemical Corporation (Osaka, Japan). Nuclease-free water was purchased from Thermo Fisher Scientific Inc. (Waltham, MA, USA). SpeedSTAR HS DNA polymerase, and RNase inhibitor were purchased from Takara Bio Inc. (Kusatsu, Japan). QIAquick PCR purification kit was purchased from QIAGEN (Venlo, The Netherlands). Further, 10× NEBuffer 3 was purchased from New England BioLabs (Ipswich, MA, USA). Carboxyl-type 96-well ELISA plates (#MS-8708F) were purchased from Sumitomo Bakelite Co., Ltd. (Tokyo, Japan). Microplate Sealing Tape Polyolefin (#9795) was purchased from 3M (Saint Paul, MN, USA).

### 2.2. Instrumentation

A Takara Dice Touch thermal cycler (Takara Bio Inc., Kusatsu, Japan) was used for the amplification of the target dsDNA and the denaturation of crRNA. A NanoDrop One microvolume spectrophotometer was used to measure the concentration of the target dsDNA. A RD-240 V polycarbonate desiccator and a LABOPORT N 820 vacuum pump were used to dry the 96-well ELISA plate surface before non-contact printing. An AD1520 aspirate/dispense system (BioDot, Inc., Irvine, CA, USA) was used for the solenoid valve-based non-contact printing of the ssDNA reporter and the Cas12–crRNA, and the diameter of the ceramic tip of the nozzle was set to 100 μm. A VP5000 series vacuum pump (BioDot, Inc., Irvine, CA, USA) was used to remove the excess droplets on the nozzle of the non-contact printer. Humidifier Ultrasonic U300 (BONECO healthy air, Widnau, Switzerland) was used to control the humidity in the printing box. A Sanyo MOV-112 (U) drying oven (Sanyo Electric Co., Ltd., Osaka, Japan) was used to control the temperature of the collateral cleavage reaction. An Olympus MVX10 macro-zoom fluorescence microscope system (Olympus Corporation, Tokyo, Japan), Olympus U-HGLGPS light source (Olympus Corporation, Tokyo, Japan), and ImagEM EM-CCD camera (Hamamatsu Photonics K.K., Hamamatsu, Japan) were used to take a fluorescence image of an enzyme-linked immunosorbent assay (ELISA) plate surface. A Synergy HTX plate reader (Agilent Technologies Inc., Santa Clara, CA, USA) was used to measure the fluorescence intensity of fluorophore-quencher (F-Q) ssDNA reporter.

### 2.3. Preparation of Target dsDNA from pEGFP-N1 and pVenus-N1 Plasmids

Initially, pEGFP-N1 and pVenus-N1 dsDNAs were amplified, and the PAM sequence (5′-TTTN-3′) was inserted into the amplified dsDNA sequence by PCR. The components of the PCR mixture are listed in Appendix A, and the following thermal cycling procedure was used: (1) 98 °C for 30 s, (2) 98 °C for 10 s, (3) 59 °C for 10 s, (4) 72 °C for 10 s (repeat (2)–(4) 40 times), (5) 72 °C for 2 min, and (6) 4 °C hold. Subsequently, the amplified dsDNA was purified using a QIAquick PCR purification kit according to the manufacturer’s instructions.

### 2.4. Immobilization Process of the ssDNA Reporter and the Cas12–crRNA on the Bottom Surface of the 96-Well ELISA Plate

The immobilization process is shown in a previous report [29]. Initially, the carboxylic acid groups of a 96-well ELISA plate surface were activated by incubation with 100 μL of 200 mM EDC/NHS in 50 mM MES-NaOH buffer (pH 6.0) for one hour. The sample was washed three times with 300 μL of water; subsequently, 100 μL of 250 nM ssDNA reporter and 12.5 nM NTA in 100 mM phosphate buffer (pH 7.5) were dropped into the well and the sample was incubated for over 12 h. After washing with 300 μL of water, the ssDNA reporter/NTA-immobilized surface was blocked by incubation with 100 μL of 1 M ethanolamine hydrochloride in 100 mM phosphate buffer (pH 7.5) for one hour. After washing with 300 μL of water, the Ni–NTA complex was formed by incubation with 100 μL of 100 mM nickel (II) chloride in water for one hour. During the incubation, crRNA was denatured at 95 °C for five minutes and cooled to 4 °C at 0.1 °C/s, and the well surface was washed two times with water. Finally, both 500 nM Cas12 and 1 μM crRNA in 1× NEBuffer 3 were applied by a micropipette or a non-contact printer. Applied solution (60 µL by a micropipette or 5–50 nL by a non-contact printer) was immobilized on the surface by incubation for two hours. Subsequently, the well surface was washed at most five times with 300 µL of NEBuffer 3 with/without shaking, as shown in Appendix A. All immobilization processes were performed at room temperature.

### 2.5. Non-Contact Printing of the Cas12–crRNA

Non-contact printing of the Cas12–crRNA was performed by using a system including the basic unit of the system (AD1520 aspirate/dispense system) and extra units, vacuum pump (VP5000 series) and humidifier (Humidifier Ultrasonic U300), according to the manufacturer’s manuals (Figure 2). After Ni–NTA formation and vacuum drying, the 96-well plate for the array substrate was set at the positions shown in Figure 2. Initially, the nozzle was moved to the reagent well by the step motor and the reagent was aspirated by depressurizing the syringe. Subsequently, the nozzle was moved onto the array substrate, and the Cas12–crRNA was dispensed by pressurizing the syringe and opening the solenoid valve in the nozzle. The non-contact printing system was operated by Axsys software ver. 2.0.9.15 (BioDot Inc., Irvine, CA, USA) (Appendix A) and the alignment of spots was entered into an Excel file ver. 2204 (Microsoft Corporation, Redmond, WA, USA).

The temperature and humidity in the printing box were maintained at approximately 25 °C and 95%, respectively, to prevent drying of the printed Cas12–crRNA droplets. When the printing volume was below 50 nL, the Cas12-printed well plate was covered with a polyolefin seal to minimize the evaporation of printed droplets.

The nozzle was cleaned inside and outside by “Priming” (the continuous flow of the system water in the nozzle) and “Washing” (inserting the nozzle into the hole in the washing area and running washing water) operations before the start of printing, during reagent change, and after Cas12–crRNA printing.

### 2.6. Incubation of Target dsDNA on the Non-Contact-Patterned SPCC-Based Sensor Array and Capturing Fluorescence Images

Target dsDNA was dissolved in 1× NEBuffer 3 with 0.5 U/μL RNase inhibitor. Subsequently, 60 µL of the dsDNA solution was dropped into the wells and incubated at 37 °C for two hours. The well was washed five times with 300 μL of water and fluorescence images (λ_ex_ = 535–555 nm and λ_em_ = 570–625 nm for HEX; λ_ex_ = 460–480 nm and λ_em_ = 495–540 nm for FAM) of the surface were obtained by microscopy and the EM-CCD camera. Subsequently, the mean value of the images was measured using Image J 1.50i software (NIH, Bethesda, MD, USA). The fluorescence intensity was obtained by subtracting and normalizing the mean value of the sample data from the blank data (the fluorescence image of the surface without any fluorophores).

### 2.7. Collateral Cleavage Activity Test of Surface-Immobilized Cas12–crRNA

Initially, 100 nM pVenus-N1 amplicon and 500 nM ssDNA reporter (HEX-Poly T–NH2–80 nt) were dissolved in 1× NEBuffer 3 with 0.5 U/μL RNase inhibitor. Subsequently, 60 µL of the mixture was dropped onto the Cas12-immobilized surface and incubated for 2 h. The fluorescence intensity of the solution was measured using a plate reader at λ_ex_ = 518–538 nm and λ_em_ = 565–575 nm. The fluorescence intensity was indicated after background subtraction and normalization using the blank solution (the mixture incubated on the surface without the Cas12–crRNA).

### 2.8. Calculating the Area of the Cas12-Immobilized Region and Its FAM-Fluorescence Intensity

The extraction process of the Cas12-immobilized region was conducted using Image J software (Appendix A). Initially, the FAM image was smoothed, and the binary image was fabricated by Otsu’s method. Second, the image was inverted, and the range of the black area was memorized by the ROI Manager. The area of the Cas12-immobilized region and its FAM-fluorescence intensity were calculated based on the extracted area in the ROI manager.

## 3. Results

### 3.1. Minimization of Surface-Immobilization Time of Cas12–crRNA Complexes

In our previous report, the SPCC-based sensor successfully distinguished two types of target dsDNA sequences in a one-pot manner [29]. However, the sizes of each spot were on the microliter (μL) scale, which proved insufficient for highly accumulated spot patterns. Therefore, the objective of this study is to pattern Cas12–crRNA complexes in nanoliter (nL)-scale droplets to develop a miniaturized and accumulated SPCC-based sensor array. However, reducing the size of the droplets used for spot creation increases the surface area per volume, making the effects of evaporation more pronounced. Consequently, Cas12–crRNA complex in non-contact printed droplets faced the risk of inactivation owing to the loss of hydration. Furthermore, the molecular motion of Cas12–crRNA during the surface-immobilization carries the risk of inactivation. Therefore, we investigated the appropriate time to immobilize Cas12–crRNA complexes before the non-contact-patterning of the SPCC-based sensor array, aiming to reduce the impact of solvent evaporation and molecular motion.

Initially, we prepared a Ni–NTA surface and dispensed 60 µL of Cas12 using a micropipette (Figure 3a). After immobilization for from 0 to 120 min, the surface was washed, and a mixed solution of 100 nM target dsDNA and 500 nM fluorophore-quencher (F-Q) ssDNA reporter was applied. Subsequently, the collateral cleavage activity was measured as the fluorescence increase in the mixed solution. The fluorescence intensity from the F-Q ssDNA reporter (indicating collateral cleavage activity) was its maximum at an immobilization time of 10 min (Figure 3b). Based on Student’s *t*-test results, there were no statistical differences in fluorescence intensity between the surface immobilization time of 10 min and that of ≥20 min. However, the fluorescence intensity gradually decreased with an increase in the immobilization time. As expected, the tendency of gradual decrease in the fluorescence is attributed to the reduction in collateral cleavage activity owing to molecular motion of the Cas12–crRNA. Although we did not investigate an immobilization time between 0 and 10 min, these conditions have limited significance for study because immobilization for under 10 min is difficult to perform when using non-contact printing owing to its mechanical property. Therefore, we selected an immobilization time of 10 min in the following studies.

### 3.2. Evaluation of Shear Stress from the Non-Contact Printing Nozzle against Surface-Immobilized Cas12–crRNA

In addition to evaporation and molecular motion, shear stress during ejection of the sensor protein from the printing nozzle is associated with its inactivation in non-contact-printed protein arrays [42,43]. Therefore, we compared the dispensing methods for Cas12–crRNA complex, non-contact printing and the micropipette, to measure the collateral cleavage of Cas12–crRNA immobilized on the surface and evaluate the effect of the shear stress.

We prepared a Ni–NTA surface and applied Cas12 using the non-contact printer (50 nL × 1200 times) or the micropipette (60 µL) (Figure 4a). After washing off the unbound Cas12–crRNA complex, the collateral cleavage activity was measured as the fluorescence increase of F-Q ssDNA reporter in the mixed solution, as described in the previous section.

The solution incubated on the surface with the Cas12–crRNA complex dispensed and immobilized using the non-contact printer exhibited an increase in fluorescence compared to the surface without Cas12 (Figure 4b). Thus, the Cas12-immobilized surface fabricated by the non-contact printer successfully showed collateral cleavage activity, although the total printing volume of the Cas12–crRNA complex solution was still large (60 µL). In addition, Student’s *t*-test of the fluorescence intensities between the “Non-contact printer” and “Micropipette” did not show significant differences. Therefore, Cas12 protein was robust against shear stress derived from the printing nozzle.

### 3.3. Optimization of the Washing Process for Immobilization of Cas12–crRNA Complex by Non-Contact Printing

Cas12–crRNA complexes immobilized for 10 min exhibited significant collateral cleavage activity and were unaffected by shear stress from the non-contact printer. Therefore, we next attempted to evaluate the pattern of the SPCC-based spot using the non-contact printer. Considering that these spots are intended to achieve multiplex dsDNA detection, the washing process for removing unbound Cas12–crRNA complex is of crucial importance. However, the washing process carries a risk of peeling off the immobilized Cas12–crRNA complex from the surface. Thus, excessive washing should be avoided. Consequently, we attempted to optimize the washing times (exchanges of washing solution) so that unbound Cas12–crRNA complex was sufficiently removed without detaching the immobilized Cas12–crRNA complex.

Following the blocking and Ni–NTA formation process, droplets including 50 nL of Cas12–crRNA complex were printed onto the surface as shown in Figure 5a, and immobilized for 10 min. In this experiment, the 3′ edge of pVenus-N1-targeting crRNA sequence was modified with FAM as shown in Figure 5b. Thus, we can evaluate the presence of the Cas12–crRNA complex and the residual amount of unbound Cas12–crRNA complex as the FAM-fluorescence intensity. Finally, the surface was washed several times, as shown in Appendix A. On the other hand, excessive washing involving physical stress induced by a shaker, as shown in Appendix A, was not adopted due to observed detachment of Cas12, as detailed in Appendix B.

After the washing process, FAM-fluorescence images and areas representing residual Cas12 and crRNA complexes were obtained, as shown in Figure 5c,d, respectively. Figure 5d shows the area of the immobilized region of the Cas12–crRNA complex. It was determined that it did not change with the number of washes, and no detachment of Cas12 was observed. Furthermore, Figure 5e indicates the FAM-fluorescence intensity under each condition. The fluorescence intensity decreased with increased washing times (up to three washes), indicating the effective removal of unbound Cas12 from the surface. Meanwhile, the decrease in fluorescence plateaued after three or more washes. Therefore, we optimized the washing times after the non-contact-printing-based immobilization of Cas12–crRNA complex to three times.

### 3.4. Miniaturization of the Non-Contact-Patterned SPCC-Based Sensor Array

We aimed to print a low volume of Cas12–crRNA and to evaluate the minimum size of the SPCC-based sensing spots for detecting target dsDNA by using non-contact printing. The SPCC-based sensor array spots were patterned (Figure 5a) by non-contact printing of 5, 10, 20, and 40 nL droplets of Cas12–crRNA complexes. After 10 min of immobilization and three washes, 60 µL of 0–100 nM pVenus-N1 dsDNA was dropped and incubated for two hours. After washing, HEX-fluorescence images were obtained.

Representative HEX-fluorescence images under various printing volume conditions ((Target dsDNA) = 0 nM) (Figure 6a) showed that the spot diameter decreased to 0.40 ± 0.11 nm with decreasing printing volumes of Cas12–crRNA complexes solution. HEX-fluorescence images after incubation with various concentrations of the target dsDNA showed that the spots gradually darkened with an increase in the target dsDNA concentration under the 40 and 20 nL printing volume conditions (Figure 6b). Conversely, the 10 nL printing volume condition maintained bright fluorescence at the edge of the spot circle while the target dsDNA was incubated. The solvent in the 10 nL droplets might be more readily evaporated compared to the solvent in the 40 and 20 nL droplets considering that the evaporation speed of a printed droplet increases with the decrease in the droplet radius [44]. In addition, the evaporation speed at the edge of the droplet is faster compared to that at the center of the droplet [45]. Thus, the cause of the bright fluorescence at the edge was thought to be the inactivation of Cas12–crRNA owing to the loss of the hydrated environment through the fast evaporation speed. Several spots were not correctly patterned at the 5 nL printing volume. We considered that the 5 nL droplets were fully dried during immobilization.

The HEX-fluorescence intensities of printing volumes ranging from 10 to 40 nL were obtained from the center part of the spot (Appendix A). The averaged HEX-fluorescence of seven spots × 4 arrays (Appendix A) was used to calculate the cleavage ratio (percentage reduction in the HEX-fluorescence from the 0 nM dsDNA condition) (Figure 6c–e). Regardless of the bright fluorescence at the edge in the 10 nL printing volume condition, the cleavage ratio for the printing volumes increased with the rise in the target dsDNA concentration from 10 to 40 nL.

Focusing on the maximum cleavage ratio, the 10 nL printing volume condition showed a lower value (74.8%) than that of the 40 and 20 nL printing volume conditions (88.8 and 91.4%, respectively). According to the coffee ring effect, molecules in the small droplets might easily accumulate at the edge of the droplet [45,46]. Thus, the lowest dynamic range being observed in the 10 nL printing volume condition might be caused by the low amount of Cas12–crRNA at the center of the spot. Considering that the coffee ring effect depends on the surface geometry, the dynamic range of the SPCC can be improved by studying the substrate material [47].

The cleavage ratio is considered to depend on the adsorbed amount of the target dsDNA on the Cas12-immobilized surface, since the immobilized ssDNA reporter in the SPCC spots does not diffuse into the liquid phase. We employed Langmuir’s adsorption isotherm model to fit the calibration plots of Figure 6c–e into the theoretical curve [48].

The adsorption rate *v_a_* and the dissociation rate *v_d_* are expressed as Equations (1) and (2), respectively, when the adsorbent (concentration in liquid: *C*) is adsorbed or desorbed on the adsorption sites (total amount: *N*).
(1)$${v_{a}=k}_{a}CN\left(1-\theta \right)$$
(2)$$v_{d}=k_{d}N\theta$$
where θ is the coverage of the adsorption sites with the adsorbent, *k_a_* is the absorption rate constant, and *k_d_* is the dissociation rate constant. In the equilibrium state (*v_a_* = *v_d_*), the dissociation constants (*K_d_*) can be calculated from Equations (1) and (2).
(3)$$K_{d}=\frac{k_{d}}{k_{a}}=\frac{CN\left(1-\theta \right)}{N\theta }$$

When the sensing signal is dependent on the amount of adsorption, the relationships among θ, the signal intensity *I*, and the maximum signal intensity *I_max_* are shown in Equation (4).
(4)$$\theta =\frac{I}{I_{max}}$$

According to Equations (3) and (4), the relationship between *I* and θ can be expressed as shown in Equation (5).
(5)$$I=\frac{I_{max}C}{K_{d}+C}$$

Therefore, the plots of Figure 6c–e might be fitted into Equation (5) when (dsDNA) and the cleavage ratio are substituted for *C* and *I*, respectively. Consequently, calibration plots were successfully fitted into Equation (5) with *R*^2^ values between 0.948 and 0.989 and *K_d_* values of 1.81, 0.983, and 4.60 nM in 40, 20, and 10 nL printing volumes of Cas12–crRNA complexes, respectively. Considering that the *K_d_* value of LbCas12a against the target dsDNA is 0.13 nM [49], our measured *K_d_* is thought to be a reasonable value.

Considering that the SPCC on the non-contact-patterned spots successfully followed Langmuir’s adsorption isotherm model, its limit of detection (LOD) for the target dsDNA can be calculated using Equation (5). Based on the 3σ at 0 nM dsDNA condition, the LOD was 572 pM, 531 pM, and 3.04 nM after printing 40, 20, and 10 nL of Cas12–crRNA complex, respectively. These LOD values were equivalent to conventional DNA microarray technologies with more complicated operation steps than those of SPCC [50,51,52]. Also, we attempted to calculate the limit of quantification (LOQ) based on the 10σ at 0 nM dsDNA condition. Although the LOQ in the 40 nL-printing volume condition was calculated to be 7.27 nM, the LOQ in the 20 and 10 nL-printing volume conditions could not be calculated because the 10σ value exceeded the maximum value of the cleavage ratio. Since the objective of our method is the identification of a target dsDNA sequence from PCR products, the ability of quantification is not important in our method.

In addition, we evaluated the detection performance of our method using correspondence tables between positive/negative samples and their test results (Appendix A). As a result, the sensitivities in every printing volume condition were 100%. On the other hand, the 40 and 20 nL-printing volume conditions indicated specificities of 92 and 67%, respectively. These false-positive results occurred in the dsDNA concentration of 0.5 nM (close to LOD). However, current PCR technologies (amplification capacity: approximately 10^11^) can easily produce a sub-µM order of dsDNA from as little as one copy/μL template [53], so our cut-off value is quite low compared to the concentration of nucleic acid amplicons. Therefore, the detection performance of our method is sufficient for practical application.

### 3.5. SPCC-Based Triple-Target dsDNA Detection on the Optimized Sensor Array

We performed triple-target dsDNA detection on the non-contact-patterned SPCC-based sensor array to evaluate the feasibility of the genotyping. Considering that the AD1520 non-contact printer cannot pattern multiple spots at the same time, the slight differences in the immobilization time of Cas12–crRNA on each spot caused differences in the fluorescence intensity of each spot (Appendix A). To correct the differences in the immobilization time, 20 nL of the Cas12 with the crRNA for the target-A, -B, and -C dsDNA sequences was patterned twice as symmetry order (Appendix A). As a control, the crRNA sequence that cannot hybridize with any target dsDNA sequences was printed at spot number 4. After washing, samples including 0 or 100 nM of the target-A, -B, and -C dsDNA sequences were dropped onto the SPCC-based sensor array and incubated for two hours. The HEX-fluorescence intensity in each crRNA sequence condition is shown in Appendix A based on the HEX-fluorescence images after the incubation of each dsDNA sample (Figure 7a). Furthermore, the cleavage ratio was calculated based on the decreased percentage compared to the HEX-fluorescence intensity of spot number 4 (control) (Figure 7b) to correct the variation in the HEX-fluorescence intensity among each array.

The cleavage ratio of every spot crRNA sequence was below 20% when the sample did not include any dsDNA sequences (no dsDNA). Meanwhile, the corresponding spot crRNA sequence indicated quite a large cleavage ratio (95.0%, 60.6%, and 60.4%, respectively when 100 nM target-A, -B, or -C dsDNA sequences were included in the sample,). These cleavage ratio values were significantly different from the other spots bound with mismatched crRNA sequences on the same array according to the Student’s *t*-test results.

Therefore, the feasibility of the one-pot genotyping was confirmed on the non-contact-patterned SPCC-based sensor array. Furthermore, the cleavage ratio increased in every spot when 100 nM target-A, -B, and -C dsDNA sequences were simultaneously included in the sample (target-ABC). Thus, the non-contact-patterned SPCC-based sensor array can be applied for genotyping in situations where the sample is expected to include multiple targets, for example, the analysis of multiple mutations in pathogen and co-infection diagnosis.

## 4. Discussion

The SPCC-based multiplex dsDNA detection was performed in the dispensing method of Cas12 by using a non-contact printer. The performance of the dsDNA response in SPCC was maintained even in the 10 nL-print-volume condition (Figure 6c–e). In this condition, the spot diameter was approximately 0.64 ± 0.05 mm. The proximal spot-patterning is possible owing to the non-contact droplet dispensing (Appendix A). The maximum diameter can be calculated at *d* + 3σ, assuming that the diameter of the spot (*d*) follows gaussian distribution. When the maximum spot-circles are proximally dispensed, the half-size of spot can be bedded in the regular triangle with *d* + 3σ of the side length. Thus, if 10 nL droplets were proximally dispensed, the spot-density is 190 spots/cm^2^. This estimated value is a 30-fold improvement over the previous SPCC-based sensor array prototype [29], and is higher than that of the multiplex dsDNA detection method using the collateral cleavage in massively-accumulated chambers (CALMEN [26], mCARMEN [27], and CLAMP [28]). Regardless of the high spot-density, the operation of our developed sensor array is much simpler than that of these previous technologies, considering that the SPCC system can be performed with only hybridization and washing steps.

In addition, the consumption of the Cas12–crRNA solution was at least 10 nL for the fabrication of one spot; this value was a 250-fold reduction from that of the previous SPCC-based sensor array prototype (2.5 µL for one spot).

Although the non-contact-patterned SPCC-based sensor array still has a drawback in the variation in fluorescence among spots (Appendix A), we expect that Cas12–crRNA printing using multiple nozzles will solve this.

## 5. Conclusions

A non-contact-printing-based patterning of SPCC-based sensing spots was developed as a miniaturized dsDNA sensor array. The nL-scale dispensing of the Cas12–crRNA droplets carries a risk of inducing inactivation; therefore, the immobilization time of Cas12–crRNA was optimized at 10 min. In addition, the washing process after immobilizing Cas12–crRNA was optimized at three-time washing to sufficiently remove unbound Cas12–crRNA.

The non-contact-patterned SPCC-based sensor array responded to the concentration of the target dsDNA following Langmuir’s adsorption isotherm after optimization. The LOD was 572 pM, 531 pM, and 3.04 nM in 40, 20, and 10 nL following the printing of Cas12–crRNA, respectively, according to the 3σ method. Therefore, the sensitivity of our developed sensor array was sufficient for clinical use. Finally, the developed sensor array identified triple dsDNA sequences in a one-pot manner without interferences among the spots.

The non-contact-patterned SPCC-based sensor array has a spot diameter of approximately 0.64 ± 0.05 mm (10 nL printing volume) which is high compared to previous related reports; therefore, it can be applied as an alternative tool to the conventional DNA microarray that requires more complicated operations than SPCC.

In the future, the non-contact-patterned SPCC-based sensor array with a wide variety of crRNA sequences will contribute to realizing point-of-care comprehensive genotyping that satisfies rapid detection with easy operation, portability, and inexpensiveness.

## Figures and Tables

**Figure 1 micromachines-15-00144-f001:**
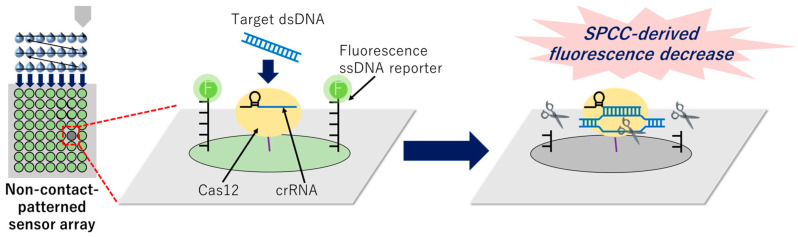
Graphical image of the miniaturized patterning of the sensor array based on non-contact printing and identification of the target dsDNA sequence by the solid-phase collateral cleavage (SPCC)-derived fluorescence decrease on a certain spot.

**Figure 2 micromachines-15-00144-f002:**
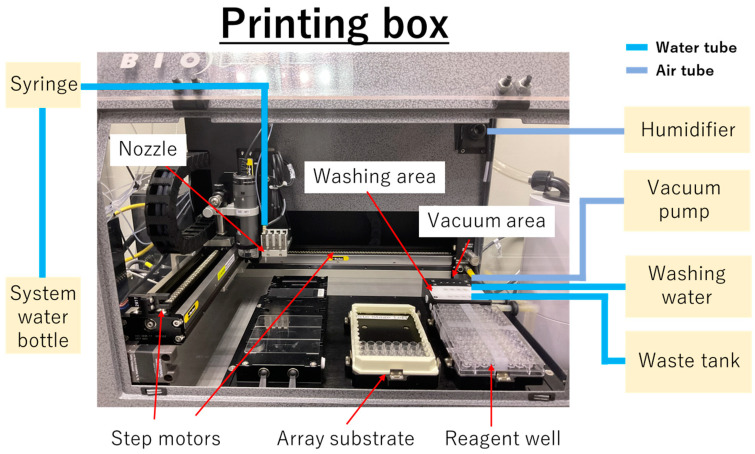
Overview of the printing box of the AD1520 aspirate/dispense system and external devices. The printing box was built on the basic unit of the system (AD1520 aspirate/dispense system) and extra units, vacuum pump (VP5000 series) and humidifier (Humidifier Ultrasonic U300). As the extra units, vacuum pump (VP5000 series) was connected to vacuum area of AD1520 aspirate/dispense system via air tube to remove residual droplets on outside of nozzle, and humidifier (Humidifier Ultrasonic U300) was connected to upper-right side of AD1520 aspirate/dispense system via air tube to maintain around 95% of the humidity inside the printing box.

**Figure 3 micromachines-15-00144-f003:**
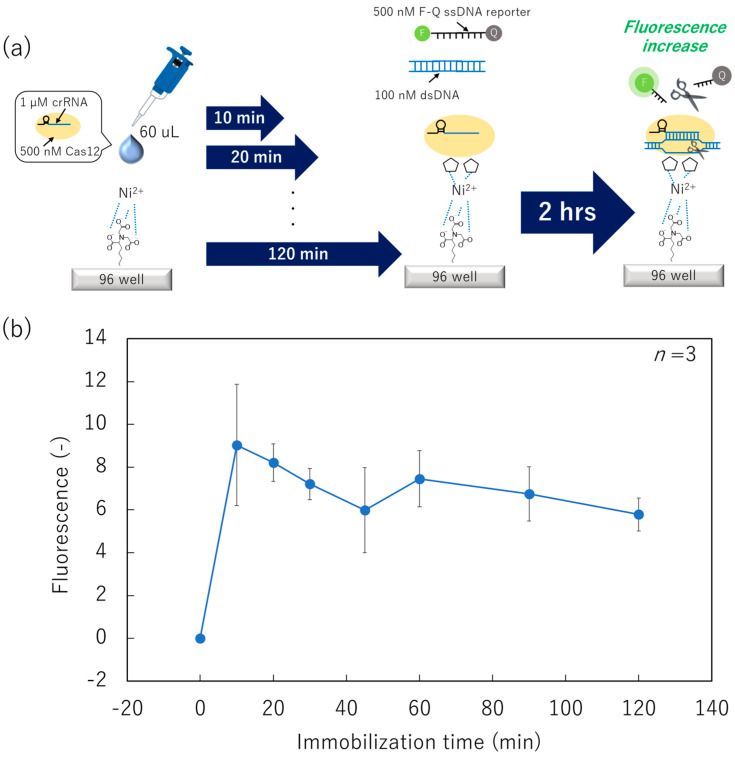
Collateral cleavage activity of surface-immobilized Cas12–crRNA in various immobilization-time conditions. (**a**) Schematic image of the dispensing/immobilization of the Cas12–crRNA and fluorescence-based collateral cleavage activity test. (**b**) Relationship between fluorescence intensity and the immobilization time of the Cas12–crRNA.

**Figure 4 micromachines-15-00144-f004:**
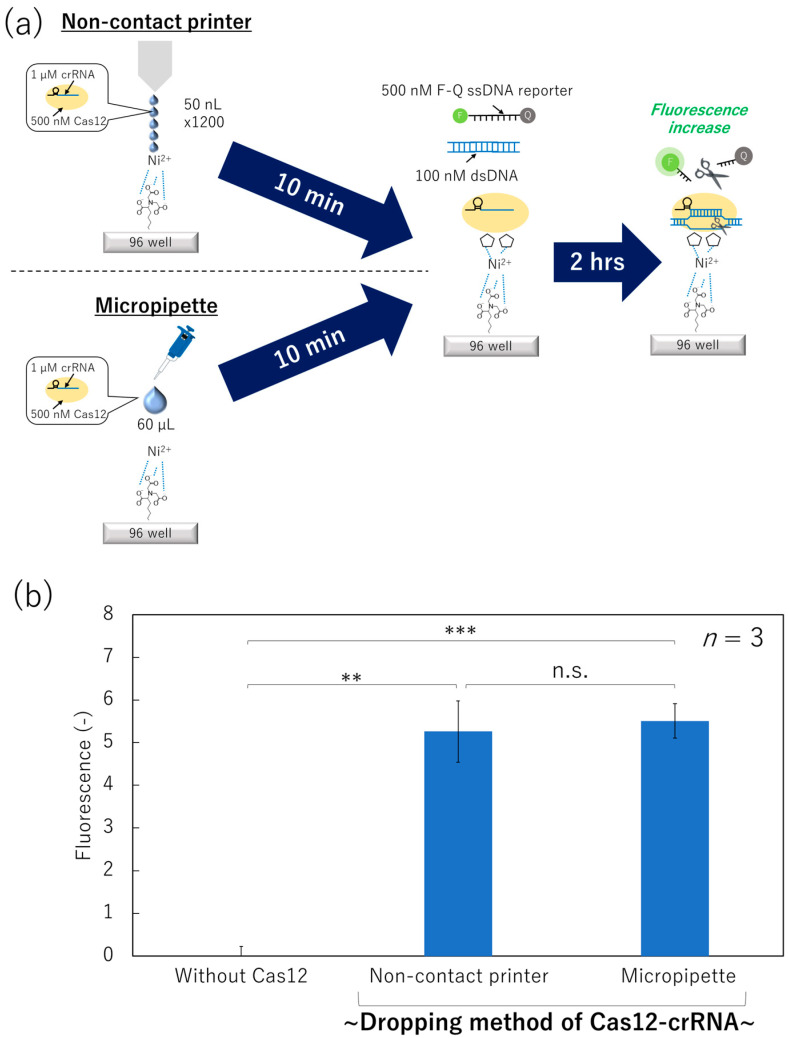
Collateral cleavage activity of the surface-immobilized Cas12–crRNA dispensed by a non-contact printer or a micropipette. (**a**) Schematic image of the dispensing/immobilization of Cas12–crRNA and the fluorescence-based collateral cleavage activity test. (**b**) Fluorescence intensity of the dsDNA/F-Q ssDNA reporter solution incubated on each surface (Student’s *t*-test result; n.s.: not significant, **: *p* < 0.005, ***: *p* < 0.0005).

**Figure 5 micromachines-15-00144-f005:**
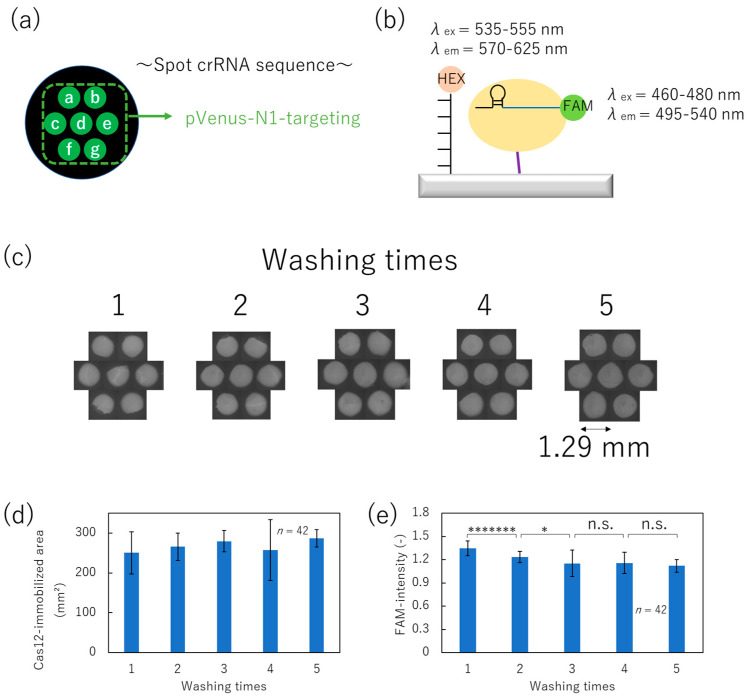
Optimization of washing times after immobilizing Cas12. (**a**) Alignment of spot with pVenus-N1-targeting crRNA. (**b**) Schematic image of HEX- and FAM-labeling. (**c**) FAM-fluorescence images of each washing condition. (**d**) Cas12-immobilized area derived from each FAM-fluorescence image. (**e**) FAM-fluorescence intensity of each condition (Student’s *t*-test result; n.s.: not significant, *: *p* < 0.05, *******: *p* < 0.00000005).

**Figure 6 micromachines-15-00144-f006:**
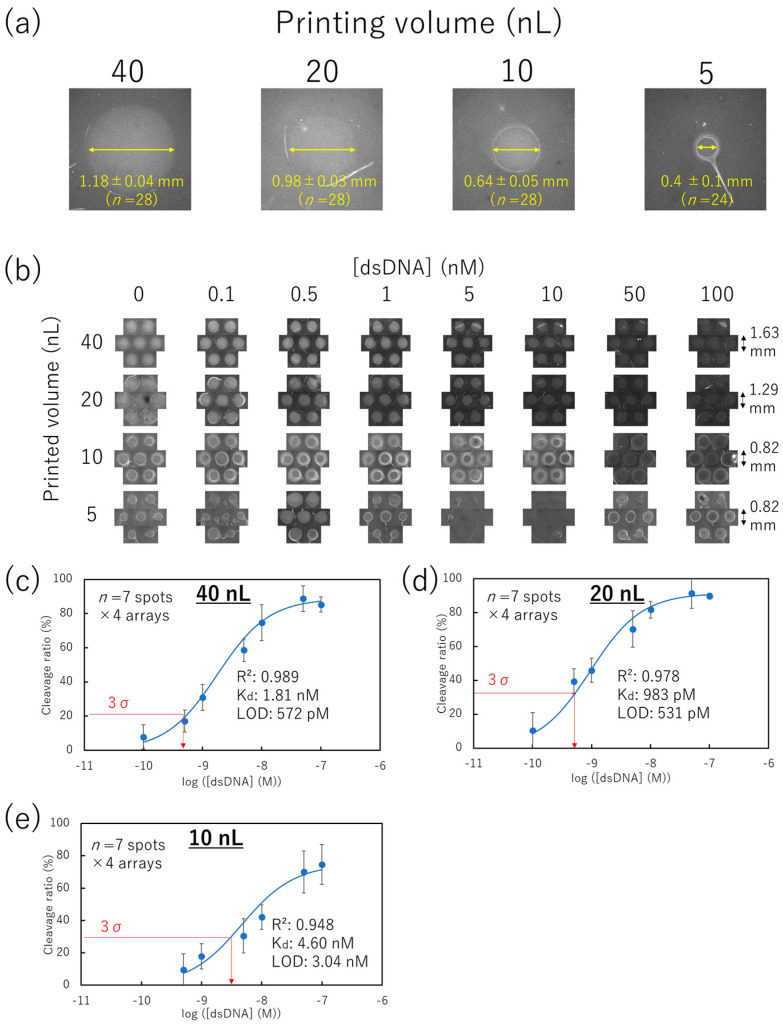
dsDNA concentration response on the non-contact-patterned SPCC-based sensor array. (**a**) Spot diameter comparison in representative HEX-fluorescence images of each printing volume condition ((Target dsDNA) = 0 nM). (**b**) HEX-fluorescence image in each printing volume of Cas12–crRNA and dsDNA concentration. (**c**–**e**) Calibration plots of the cleavage ratio at yellow circle areas at 40, 20, and 10 nL-printing volumes, respectively.

**Figure 7 micromachines-15-00144-f007:**
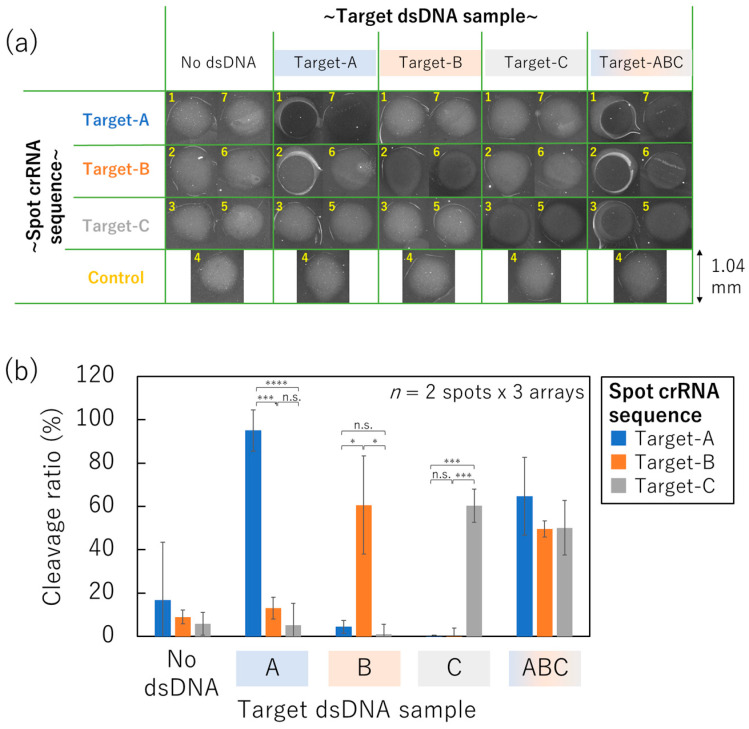
One-pot triple-target dsDNA detection on the non-contact-patterned SPCC-based sensor array fabricated by the optimized process. (**a**) HEX-fluorescence image of each spot. (**b**) Cleavage ratio in each crRNA sequence condition after incubation of each dsDNA sample (Student’s *t*-test result; n.s.: not significant, *: *p* < 5 × 10^−2^, ***: *p* < 5 × 10^−6^, ****: *p* < 5 × 10^−8^).

## Data Availability

Data are contained within the article and Appendix A.

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
