# Peer review of "Miniaturization of CRISPR/Cas12-Based DNA Sensor Array by Non-Contact Printing"

_micromachines, 2024, doi:10.3390/mi15010144_

Round 1

Reviewer 1 Report

Comments and Suggestions for Authors

The content of this manuscript is suitable to the scope of micromachines. However, the following revisions are needed for publication.

I cannot find the time dependency in Figure 3. Is there a statistically significant difference?

LOQ should also be calculated and shown.

Does the viscosity of the sample affect to this analytical method? Biological samples such as saliva and nasopharyngeal swab are highly viscous. Authors should investigate the effect of viscosity.

Author Response

Comment 1 "I cannot find the time dependency in Figure 3. Is there a statistically significant difference?"

Thank you for your nice comments. Based on your advice, we attempted the student’s t-test in fluorescence differences between the surface immobilization time of 10 min and that of ≧20 min, but there no statistical difference. However, it is obvious that fluorescence intensity was gradually decreased after 10 min. Thus, we set the surface immobilization time to 10 min.

To add the statical examination about time dependency, we modified the sentence at line 294-299 as shown below. Please refer to it.

The fluorescence intensity from the F-Q ssDNA reporter (indicating collateral cleavage activity) was maximum at an immobilization time of 10 min (Figure 3b). Based on the student’s t-test results, there were no statistical difference in fluorescence intensity between the surface immobilization time of 10 min and that of ≧20 min. However, the fluorescence intensity gradually decreased with an increase in immobilization time.

Also, to clarify the gradual fluorescence decrease, we add polygonal line in Figure 3b.

Comment 2 "LOQ should also be calculated and shown."

Thank you for your nice comment. However, the objective of this manuscript is the identification of target dsDNA sequence from PCR products as described at line 447-452. Since target dsDNA is pre-amplified, the ability of quantification is not important in our manuscript. Rather than that, since it is important whether dsDNA at the concentration level of PCR product can be detected, we only denoted the LOD.

When we calculated LOD again, we found the mistake in the LOD of 10 nL printing. Thus, we revised the value from 3.39 nM to 3.04 nM(as green color at line 26, 446, and 529). However, since the LOD was slightly improved, we consider that this revision does not affect on the story of the manuscript.

Comment 3 "Does the viscosity of the sample affect to this analytical method? Biological samples such as saliva and nasopharyngeal swab are highly viscous. Authors should investigate the effect of viscosity."

Thank you for your constructive question. As mentioned above, our purpose is the genotyping of the nucleic acid amplicons such as PCR product. Thus, the viscosity of saliva and nasopharyngeal swab will not affect on our SPCC system. According to previous reports, Cas12-based collateral cleavage system is functional under the environment of PCR and RPA buffer (ACS Synth. Biol. 2022, 11, 2, 835–842; Lab Chip, 2021, 21, 2730-2737). Thus, our system is expected to be applied in practical situations.

Reviewer 2 Report

Comments and Suggestions for Authors

This work introduces an interesting detection system of CRISPR/Cas12-based DNA sensor array by non-contact printing. The experiments are designed well and the results support the conclusion. The English writing needs slight improvement.

The authors try to analyse the relationship between properties of spots and the results. I suggest the authors can read and cite the two papers below, which may be helpful:

1. "Protein microarray spots are modulated by patterning method, surface chemistry and processing conditions", Biosensors and Bioelectronics, 2019.

2. ''The Influence of Substrate Microstructures on the Fluorescent Intensity Profile, Size, Roundness, and Coffee Ring Ratio of Protein Microarray Spots'', Proceedings of IEEE NEMS 2022, 2022.

In addition, for some figures (Figure 4, Figure 5...): it seems there exists many frames of subfigures which should be removed, please double check. 

Comments on the Quality of English Language

1. Line 29: 'micro-array' should be 'microarray'

2. Line 43-44: 'several days of pathogen culture' should be 'pathogen culture of several days'

3. Line 132: 'micro-pattern' should be 'micropattern'

4. Line 362: 'that' should be removed.

5. Line 441: '40-, 20-' should be '40, 20'

6. Line 483-485: should be 'Thus, the non-contact-patterned SPCC-based sensor array can be applied for genotyping in situations where the sample is expected to include multiple targets, for example, the analysis of multiple mutations in pathogen and co-infection diagnosis.'

7. Line 495-496: 'The SPCC-based multiplex dsDNA detection performed in the dispensing method of Cas12 by using a non-contact printer' needs to be rephrased. 

8. Line 522: 'times' should be 'time'

9. Line 525: 'Langumuir' should be 'Langmuir'

Author Response

Comment 1.

 The authors try to analyse the relationship between properties of spots and the results. I suggest the authors can read and cite the two papers below, which may be helpful:

・"Protein microarray spots are modulated by patterning method, surface chemistry and processing conditions", Biosensors and Bioelectronics, 2019.

・''The Influence of Substrate Microstructures on the Fluorescent Intensity Profile, Size, Roundness, and Coffee Ring Ratio of Protein Microarray Spots'', Proceedings of IEEE NEMS 2022, 2022.

Answer: Thank you for your useful advice. Since Biosensors and Bioelectronics, 2019 describes the phenomenon of the coffee ring effect in protein microarray fabrication, we cite it at the line 414 as reference number of 47. In addition, since Proceedings of IEEE NEMS 2022 compared the coffee ring effect on different material, we add below sentence at line 416-418 and cited this paper as reference number of 48.

Considering that the coffee ring effect depends on the surface geometry, the dynamic range of the SPCC can be improved by the study of the substrate material [48].”

Please refer to the revised manuscript.

Comment 2. In addition, for some figures (Figure 4, Figure 5...): it seems there exists many frames of subfigures which should be removed, please double check.

Answer: Thank you for your comments. However, we could not find frames of subfigures. Probably because the background of graph in previous manuscript was white, frames might be appeared in the high-resolution display. To remove this risk, we made the background of every graph transparent. Would you check whether frames are successfully removed?

Comment 3: Comments on the Quality of English Language

  1. Line 29: 'micro-array' should be 'microarray'
  2. Line 43-44: 'several days of pathogen culture' should be 'pathogen culture of several days'
  3. Line 132: 'micro-pattern' should be 'micropattern'
  4. Line 362: 'that' should be removed.
  5. Line 441: '40-, 20-' should be '40, 20'
  6. Line 483-485: should be 'Thus, the non-contact-patterned SPCC-based sensor array can be applied for genotyping in situations where the sample is expected to include multiple targets, for example, the analysis of multiple mutations in pathogen and co-infection diagnosis.'
  7. Line 495-496: 'The SPCC-based multiplex dsDNA detection performed in the dispensing method of Cas12 by using a non-contact printer' needs to be rephrased.
  8. Line 522: 'times' should be 'time'
  9. Line 525: 'Langumuir' should be 'Langmuir'

Answer: Thank you for your advice. We revised as red letters. Please refer to the revised manuscript.

Round 2

Reviewer 1 Report

Comments and Suggestions for Authors

I understanded the authors' comment about LOD, and LOQ. However, maybe, LOQ cannot be calculated, because the value of 10σ would exceed 100% cleavage ratio. Authors mentioned that the ability of quantification is not important, but I think that this analytical method has large errors and low sensitivity, with a 10σ greater than 100% cleavage ratio, and there are going to be many false positives. If the concentration of dsDNA below the LOD is used as the negative sample and fitted to the calibration curve, what is the ratio of negative (values below 3σ) to positive (values above 3σ)? Since n = 7, please indicate using that value. The results should also be mentioned in discussion.

Author Response

Comment: I understanded the authors' comment about LOD, and LOQ. However, maybe, LOQ cannot be calculated, because the value of 10σ would exceed 100% cleavage ratio. Authors mentioned that the ability of quantification is not important, but I think that this analytical method has large errors and low sensitivity, with a 10σ greater than 100% cleavage ratio, and there are going to be many false positives. If the concentration of dsDNA below the LOD is used as the negative sample and fitted to the calibration curve, what is the ratio of negative (values below 3σ) to positive (values above 3σ)? Since n = 7, please indicate using that value. The results should also be mentioned in discussion.

Answer: Thank you for your constructive opinion. As you pointed out, LOQ cannot be calculated in some cases because the value of 10σ exceeds a 100% cleavage ratio. Consequently, we made correspondence tables between positive and negative samples and their respective test results (Table S4) to assess whether our method’s detection performance is suitable for practical applications. The outcomes revealed 100% sensitivity under all printing volume conditions. However, under 40 and 20 nL-printing volume conditions, specificities were 92 and 67%, respectively. These instances of false-positive results occurred at a dsDNA concentration of 0.5 nM, close to LOD.

Neverthless, it is crusial to note that the non-contact-patterned SPCC-based sensor array is designed for genotyping PCR products. Current PCR technologies, with an amplification capacity of  approximately 1011, can effortless generate sub-µM concentrations of dsDNA from as little as 1 copy/μL of the template. Given that our cut off value is significantly lower than the concentration of nucleic acid amplicons, the detection performance of our method proves to be sufficient for practical applications. Additionally, we confirmed that non-contact-patterned SPCC-based spots specifically detect sub-µM dsDNA with sequences complementary to their crRNA, as shown in Figure 7.

To incorporate your comment and our response into our manuscript, we have revised the sentences from line 448 to 464, as shown below, and included Table S4 in the supporting information. Please refer to these revisions.

“Also, we attempted to calculate limit of quantification (LOQ) based on 10σ at 0 nM dsDNA condition. Although LOQ in 40 nL-printing volume condition was calculated to be 7.27 nM, LOQ in 20 and 10 nL-printing volume conditions could not be calculated because the 10σ value exceeded the maximum value of the cleavage ratio. Since the objective of our method is the identification of target dsDNA sequence from PCR products, the ability of quantification is not important in our method.

In addition, we evaluated the detection performance of our method from correspondence tables between positive/negative samples and their test results (Table S4). As a result, sensitivities in every printing volume condition are 100%. On the other hands, 40 and 20 nL-printing volume conditions indicated specificities of 92 and 67%, respectively. These false-positive results were occurred in the dsDNA concentration of 0.5 nM (close to LOD). However, current PCR technologies (amplification capacity: approximately 1011) can easily produce sub-µM order of dsDNA from as little as 1 copy/μL template [54], so our cut-off value is quite lower compared to the concentration of nucleic acid amplicons. Therefore, the detection performance of our method is sufficient for the practical application.”

Round 3

Reviewer 1 Report

Comments and Suggestions for Authors

Authors revised according to reviewer comment. I recommend publishing of this manuscript.